# Improved Grey Wolf Optimization Algorithm and Application

**DOI:** 10.3390/s22103810

**Published:** 2022-05-17

**Authors:** Yuxiang Hou, Huanbing Gao, Zijian Wang, Chuansheng Du

**Affiliations:** 1School of Information and Electrical Engineering, Shandong Jianzhu University, Jinan 250101, China; 2020080118@stu.sdjzu.edu.cn (Y.H.); zijian1632022@163.com (Z.W.); chuansheng2022@163.com (C.D.); 2Shandong Key Laboratory of Intelligent Building Technology, Jinan 250101, China

**Keywords:** Grey Wolf Optimizer, tent mapping, convergence factor, path planning

## Abstract

This paper proposed an improved Grey Wolf Optimizer (GWO) to resolve the problem of instability and convergence accuracy when GWO is used as a meta-heuristic algorithm with strong optimal search capability in the path planning for mobile robots. We improved chaotic tent mapping to initialize the wolves to enhance the global search ability and used a nonlinear convergence factor based on the Gaussian distribution change curve to balance the global and local searchability. In addition, an improved dynamic proportional weighting strategy is proposed that can update the positions of grey wolves so that the convergence of this algorithm can be accelerated. The proposed improved GWO algorithm results are compared with the other eight algorithms through several benchmark function test experiments and path planning experiments. The experimental results show that the improved GWO has higher accuracy and faster convergence speed.

## 1. Introduction

Path planning is widely used in mobile robot navigation, which of the aim is to find an optimal trajectory that connects the starting point with the target point while avoiding collisions with obstacles [1,2]. There are many commonly used algorithms, such as A* algorithm [3], particle swarm algorithm (PSO) [4,5], genetic algorithm (GA) [6], and grey wolf algorithm (GWO) [7,8,9].

GWO is a new pack intelligence optimization algorithm that is widely used in many significant fields. It mainly imitates the grey wolf race pack’s hierarchical pattern and hunting behavior and achieves optimization through the wolf pack’s tracking, encircling, and pouncing behaviors. Compared with traditional optimization algorithms such as PSO and GA, GWO has the advantages of fewer parameters, simple principles, and implementing easily. However, GWO has the disadvantages of slow convergence speed, low solution accuracy, and easy to fall into the local optimum. For this reason, many scholars have made many improvements. Yang Zhang [10] proposed MGWO, which introduced an exponential regular convergence factor strategy, an adaptive update strategy, and a dynamic weighting strategy to improve the GWO search capability. Min Wang [11] proposed NGWO, which used reverse learning of the initial racial group and introduced a nonlinear convergence factor to improve the algorithm search capability. Luis Rodriguez [12] proposed the Grey Wolf algorithm (GWO-fuzzy) based on a fuzzy hierarchical operator and compared two proportional weighting strategies. Saremi [13] proposed the grey Wolf Algorithm for Evolutionary Population Dynamics (GWO-EPD), which focuses on the location change of poorly adapted grey wolf individuals to improve search accuracy. Qiuping Wang [14] proposed an improved grey wolf algorithm (CGWO), which uses the cosine law to vary the convergence factor to improve the searchability, and introduces a proportional weight based on the step Euclidean distance to update the position of the grey wolf to speed up the convergence speed. Shipeng Wang [15] proposed a new hybrid algorithm (FWGWO), which combines the advantages of both algorithms and effectively achieves the global optimum. In order to effectively improve the coverage of a wireless sensor network in the monitoring area, a coverage optimization algorithm for wireless sensor networks with a Virtual Force-Lévy-embedded Grey Wolf Optimization (VFLGWO) algorithm is proposed [16].

Although the GWO algorithm has been widely used in various engineering problems, such as numerical simulation and stability domains [17,18], classification of data sets, feature acquiring selection, etc., it has been less applied in mobile robot path planning. The research object is the path planning of mobile robots. The shortest path is the objective function, the environment is the constraint condition, and the grey wolf optimization algorithm applies to the path planning of mobile robots to avoid obstacles. To address the defects of the gray wolf optimization algorithm in solving the path planning problem of mobile robots, such as falling into local extremes, poor stability, and poor local search capability. Summarizing the above research results, we know that there are three factors determining the performance of the grey wolf algorithm in finding the best path: the initialized wolf pack, the convergence factor, and the proportional weighting strategy. In this paper, we mainly improve these three aspects of GWO. First, initialize the wolf pack position using improved chaotic tent mapping. The second is applying a nonlinear convergence factor based on the Gaussian distribution variation to improve the search capability. Finally, a dynamic weighting strategy is introduced to speed up the convergence. Several benchmark functions are simulated and compared with various improved GWO and classical intelligent optimization algorithms to show the effectiveness of the improved algorithms. The improved GWO has been tested on mobile robot path planning to verify the algorithm’s practicality.

The contributions of this paper are:An improved GWO algorithm based on a multi-strategy hybrid is proposed.The improved GWO algorithm is applied to the path planning of mobile robot.The performance of the proposed approach is compared with standard GWO, Sparrow Search Algorithm (SSA), Mayfly Algorithm (MA), Modified Grey Wolf Optimization Algorithm (MGWO) [10], Novel Grey Wolf Optimization Algorithm (NGWO) [11], A Fuzzy Hierarchical Operator in the Grey Wolf Optimizer Algorithm (GWO-fuzzy) [12], and Evolutionary population dynamics and grey wolf optimizer (GWO-EPD) [13].

The remainder of this paper is structured as follows. Section 2 summarizes the related work. Section 3 describes the deployment scheme of this paper to improve the gray wolf algorithm. The experimental results are discussed in Section 4. Section 5 concludes the paper.

## 2. Related Work

### 2.1. Research Situation

Path planning is a typical complex multi-aim optimization problem that finds a workable or optimal path from the starting point to the goal point under careful consideration of various environmental conditions. Intelligent algorithms are widely used in such problems as path planning because of their better robustness.

Research on solving path planning problems using swarm intelligence algorithms is gradually increasing. For example, Yin Ling [19] fused the improved grey wolf algorithm with the artificial potential field method to solve the problem of unreachable target points because of the influence of dynamic obstacles in path planning. Dazhang You [20] combined GWO with particle swarm algorithm to reduce the cost consumption of path planning by introducing cooperative quantitative optimization of the grey wolf population. Kumar R [21] introduced a new technique named modified grey wolf optimization (MGWO) algorithm to solve the path planning problem for multi-robots. Ge Fawei [22] proposed the grey wolf fruit fly optimization algorithm (GWFOA), which combines the fruit fly optimization algorithm (FOA) with GWO for the Unmanned Aerial Vehicle (UAV) path planning problem in oil field inspection, resulting in a satisfactory solution for UAV in complex environments. One more powerful algorithm named variable weight grey wolf optimization (VW-GWO) was recently proposed by Kumar [23] to obtain an optimal solution for the path planning problem of mobile robots.

### 2.2. GWO Algorithm

In 2014, inspired by the predatory behavior of grey wolf packs, Seyedali Mirjalili et al. proposed the grey wolf algorithm (GWO) [7]. The algorithm simulates the unique hunting and prey-seeking characteristics of the grey wolf. Grey wolves belong to the group of living canines. Each wolf plays a different role in the group and accomplishes tasks through cooperation between wolves. The GWO divided the grey wolf population into four levels of social hierarchy (Figure 1). The first rank is *wolf α*, responsible for deciding on activities such as hunting. The second rank is *wolf β*, subordinate to *wolf α* and helps make decisions with *wolf α*, also the best candidate for wolf α. The third rank is *wolf δ*, subordinate to *wolf α* and *wolf β*, responsible for tasks such as scouting and hunting. The fourth rank is *wolf*
*ω*, the lowest rank, responsible for maintaining the wolf pack. Grey wolf hunting is divided into tracking, chasing, and attacking prey.

During the GWO operation, the positions of *wolf α*, *wolf β*, and *wolf δ* are continuously updated at each iteration, whose mathematical model is described as:(1)$$D=|C•X_{p}(t)-X(t)|\omega$$
(2)$$X(t+1)=X_{p}(t)-A•D$$

Equation (1) is the distance between the grey wolf and the prey, where *t* is the number of current iterations, and *X_p_*(*t*) and *X*(*t*) are the prey’s locations and the grey wolf’s location at t iterations, respectively. Equation (2) is the formula for updating the location of the grey wolf. *A* and *C* are the coefficient vectors, which are calculated by the following equations:(3)$$A=2a•r_{1}-a$$
(4)$$C=2•r_{2}$$
where *r*_1_, *r*_2_ are random vectors between [0, 1], and the primary role is to increase the randomness of the grey wolf movement. *a* represents the convergence factor, which will decay linearly from 2 to 0 as the algorithm progresses, and the linear relationship defines GWO:(5)$$a=2-2t/T_{\max}^{}$$
where *t* is the current number of iterations and *T*_max_ is the maximum number of iterations of the algorithm.

Predating in abstract space and accurately identifying the location of prey is impossible. GWO simulated hunting behavior. Based on the fitness value, *wolf*
*α*, *wolf*
*β*, and *wolf δ* were selected to find the prey using the relationship between the three positions and guide the other wolves to move toward the prey, as in Figure 2.

By iterating several times until the location of the prey is reached, the mathematical model is as follows:(6)$$\left\{\begin{array}{c} D_{a}=|C_{1}•X_{\alpha }-X|\\ D_{\beta }=|C_{2}•X_{\beta }-X|\\ D_{\delta }=|C_{3}•X_{\delta }-X| \end{array}\right.$$
(7)$$\left\{\begin{array}{c} X_{1}=|X_{\alpha }-A_{1}•D_{\alpha }|\\ X_{2}=|X_{\beta }-A_{2}•D_{\beta }|\\ X_{3}=|X_{\delta }-A_{3}•D_{\delta }| \end{array}\right.$$
(8)$$X(t+1)=(X_{1}+X_{2}+X_{3})/3$$
where: *D_a_* is the distance between wolf pack *w* and *a* wolf, *D_B_* is the distance between wolf pack *w* and *β* wolf, and *D**_δ_* is the distance between wolf pack *w* and wolf *δ*. The Equation (7) presents the location of the new generation of wolves after the update.

## 3. Improved GWO Algorithm

### 3.1. Wolf Pack Initialization

Since the initialized grey wolf population determines whether the optimal path can be found and the convergence speed, a diversity of initialized populations can help improve the algorithm’s performance in finding the optimal path. Traditional GWO randomly initializes wolf pack positions, which primarily affects the search efficiency of the algorithm, so the initialized populations need to be distributed as evenly as possible in the initial space.

In optimization, chaotic mappings positively impact the convergence speed of GWO algorithms, and chaotic sequences have characteristics such as nonlinearity, ergodicity, and preventing algorithms from falling into local optimality. In the last decade, chaotic mapping has been widely used to help optimize more dynamic and global search spaces for intelligent algorithms. There are over ten mappings: logistic mapping, piecewise-linear chaotic system mapping(pwlcm), singer mapping, and tent mapping. These mappings can choose the initial value of any number [0, 1] (or according to the chaotic mapping range). Among them, logistic mapping and tent mapping are most commonly used, but logistic mapping is less ergodic than tent mapping, and the sensitivity of initial parameters leads to the high density of mapped points at the edges and less density in the middle region, which is not conducive to optimal path planning. Compared with logistic mapping, tent mapping is more suitable for GWO, but it is a small period. Therefore, a random variable *rand*()/*N* is added to the tent mapping.
(9)$$y_{i,j+1}=\left\{\begin{array}{c} \upsilon •y_{i,j}+rand()/N,0\leq y_{i,j+1}\leq 0.5\\ \upsilon •(1-y_{i,j})+rand()/N,0.5<y_{i,j+1}\leq 1 \end{array}\right.$$
where: *i* is the grey wolf pack size, *j* is the chaotic sequence number, *rand*() belongs to [0, 1], *v* belongs to [0, 2], *N* is the population number. Introducing *rand*()/*N* can maintain the ergodicity and regularity of tent mapping and effectively solve the tent falling into small and unstable periodic points during iteration. Figure 3 shows the change curves of two Tent chaotic mappings. The tent mapping has significantly improved reversibility and uniform distribution compared with the tent. Improved tent mapping steps:Produce random initial values *y*_0_ in (0, 1) with *i* = 0.Calculate iteratively using Equation (9) to produce the sequence.Stop iterating when the iteration reaches the maximum value and saves the sequence.

Finally, map it to the grey Wolf Pack search space.
(10)$$x_{i,j}=lb+y_{i,j}•(ub-lb)$$
where *lb* and *ub* are the upper and lower limits of the grey wolf position, respectively, introducing random variables in the tent mapping can effectively avoid the shortage of minor cycle points and limit the random values to a set range. Improving tent mapping enables the GWO initialized wolf pack positions to be uniformly distributed in the search space.

### 3.2. Nonlinear Convergence Factor

In GWO, the excellent or lousy convergence factor affects the algorithm’s global search ability and local exploitation ability. The global search ability is the search of the grey wolf pack to other unopened areas to prevent the wolf pack from falling into local optimal solutions. Equation (3) |*A*| > 1, the grey wolf pack needs to search the prey in the entire space. The local exploitation ability represents the accuracy in a small area. When |*A*| < 1, the grey wolf pack wants to surround and attack the prey, and the local ability also determines the convergence speed, so the convergence factor has a significant role. The convergence factor used in traditional GWO is a linear decreasing factor, decreasing from 2 to 0. However, it is found that the actual is not a linear change, and nonlinearity is more applicable to GWO. in addition, the first stage of GWO is mainly for a global search for optimal solutions, and the middle and later stages are for local development, with different needs for convergence factors.

Therefore, this paper uses a convergence factor based on the Gaussian distribution change curve.
(11)$$\left\{\begin{array}{c} a=\phi •\frac{1}{\sqrt{2\pi }(T_{\max}/3)}e^{\frac{t^{2}}{2{\left(T_{\max}/3\right)}^{2}}},t\leq \partial T_{\max}\\ a=\varphi •\frac{1}{\sqrt{2\pi }(T_{\max}/3)}e^{\frac{t^{2}}{2{\left(T_{\max}/3\right)}^{2}}},\partial T_{\max}\leq t<T_{\max} \end{array}\right.$$
where *Ø*, *φ* is the decreasing function, changes with the number of iterations, and ∂ is the cut-off. Figure 4 compares the convergence factors of GWO, Improved Gray Wolf Optimizer Algorithm (MGWO) in literature [10], and improved GWO proposed in this paper.

The convergence factor of GWO is linearly decreasing, which does not apply to the application of the algorithm in practice. The convergence factor of MGWO is based on the exponential law, which does not guarantee the accuracy of the local search at the late stage of the search. The improved convergence factor is a curve decaying according to the nonlinear normal distribution, and the convergence factor is more significant and decays slower at the beginning of the iteration so that the population can better search for the optimal solution to the unknown global region, thus improving the global searchability in the early stage and preventing it from falling into the local optimum. The convergence factor is more minor and decays more at the later iteration stage to improve the algorithm’s local search accuracy and convergence speed. The convergence factor is more minor and decays more in the later iterations, thus improving the local search accuracy and convergence speed. Therefore, the improved convergence factor can better balance GWO global search and local search ability.

### 3.3. Dynamic Proportional Weighting Strategy

The traditional GWO uses Equation (8) as the formula for wolf position update, but the effect is not good. The [24] proposed two methods to improve the position update formula by increasing the weights.
(12)$$X(t+1)=\frac{5X_{1}+3X_{2}+2X_{3}}{10}$$
(13)$$\begin{array}{l} W_{a}=\frac{f_{a}\;+\;f_{\beta }\;+\;f_{\omega }}{f_{a}},W_{\beta }=\frac{f_{a}\;+\;f_{\beta }\;+\;f_{\omega }}{f_{\beta }},\\ W_{\omega }=\frac{f_{a}\;+\;f_{\beta }\;+\;f_{\omega }}{f_{\omega }}\\ X(t\;+\;1)=\frac{X_{1}•W_{a}\;+\;X_{2}•W_{\beta }\;+\;X_{3}•W_{\omega }}{W_{a}\;+\;W_{\beta }\;+\;W_{\omega }} \end{array}$$

Equations (12) and (13) set *α*, *β*, and *w* with different coefficients to highlight their importance, and Equation (12) increases the coefficient 5 for *α*, 3 for *β*, and 2 for w according to the importance. *W* in Equation (13) denotes the weight of the three wolves, and f denotes the current adaptation degree of the three wolves and increases the weight of the wolves according to the adaptation degree.

Inspired by the above, a proportional weighting strategy based on fitness and location is proposed to make the grey wolf pack find the optimal solution more precisely:(14)$$\begin{array}{l} W_{a}=\frac{f_{a}\;+\;f_{\beta }\;+\;f_{\omega }}{f_{a}},W_{\beta }=\frac{f_{a}\;+\;f_{\beta }\;+\;f_{\omega }}{f_{\beta }},\\ W_{\omega }=\frac{f_{a}\;+\;f_{\beta }\;+\;f_{\omega }}{f_{\omega }}\\ V_{1}=\frac{\left|X_{1}|\;+\;|X_{2}|\;+\;|X_{3}\right|}{\left|X_{1}\right|},V_{2}=\frac{\left|X_{1}|\;+\;|X_{2}|\;+\;|X_{3}\right|}{\left|X_{2}\right|},\\ V_{3}=\frac{\left|X_{1}|\;+\;|X_{2}|\;+\;|X_{3}\right|}{\left|X_{3}\right|},\\ X(t\;+\;1)=\frac{V_{1}•W_{a}\;+\;V_{2}•W_{\beta }\;+\;V_{3}•W_{\omega }}{3} \end{array}$$

The complexity of the traditional GWO algorithm is O (N × d × *T*_max_). The complexity of the GWO-EPD algorithm is O (2N × d × *T*_max_), which is mainly between GWO and EPD. the complexity of the NGWO algorithm is O (3N × d × *T*_max_). The complexity of the MGWO algorithm is O (N × d × *T*_max_), which shows the number of subgroups in the operation process. The improved GWO algorithm of this paper uses chaotic tent mapping, which is based on the nonlinear convergence factor of the normal distribution, and the complexity of this algorithm is O (N2 × d × *T*_max_). The algorithm complexity shows that the algorithm complexity of the improved GWO is higher, but the comparison of the above benchmark test function shows that the solution accuracy and convergence speed are better than the other algorithms.

The improved GWO algorithm pseudo-code is shown in Algorithm 1.
**Algorithm 1:** Pseudo Code of Improved GWO1**Initialize** (*Xi* (*i* = 1, 2…, *n*)) *t*, *T*_max_, *a*,* A*,* C*2**Initialize** Tent map x_0_3**Calculate the** fitness of each wolf4  *X_a_* = best wolf. *X**_β_* = second wolf. *X_w_* = third wolf.5**While** t < *T*_max_6   **Sort**  fitness of each wolf7   **Update** chaotic number, a8   **for** each search agent9        **Update** position current wolf using10    **end**
11    **Calculate**  fitness of each wolf12    **Update** *X_a_*,* X**_β_*,* X_w_*13    *t* = *t* + 114 **end**


## 4. Result

In order to verify the performance of the improved algorithm, 15 international standard benchmark test functions are selected for simulation experiments. For the fairness of the results, the relevant parameters of all compared algorithms are configured in Table 1 and Table 2 shows the benchmark test functions. GWO, MGWO [10], NGWO [11], GWO-fuzzy [12], GWO-EPD [13], and the improved GWO in this paper were selected for comparison of simulation experiments. Simulation experiments were conducted using Matlab on a Lenovo R7000P, containing a 2020H, 2.90 GHz processor. Table 3 shows the comparison of the mean and standard deviation of the results of 30 independent runs of the algorithms, and the best results of the compared algorithms are in bold in the Table 3 and Table 4. Furthermore, Figure 5 shows the convergence curves of the six algorithms on some of the tested functions.

### 4.1. Comparison with GWO and Other Improvement GWO

#### 4.1.1. Convergence Accuracy Analysis

From the traditional GWO principle, it is known that the exploration ability of the algorithm depends mainly on the convergence factor, and in practical experiments, it can be observed that the convergence factor decays not linearly from 2 to 0 but with the number of iterations [10]. MGWO convergence factor uses a nonlinear exponential convergence factor, which will work well compared to the linear convergence factor, which illustrates the effectiveness of a nonlinear convergence factor.

The results in Table 3 show that the improved GWO algorithm outperforms several other improved algorithms tested under 15 sets of test functions because the initial set number of iterations is satisfied. The single-peak test function is mainly used to test the development capability of the algorithm. For f1, f2, f3, and f4, it can be found the theoretical optimal value of 0 in terms of the stability of the search and the accuracy of the search. In solving f7, although the effect is not very obvious after using the improved algorithm, the mean and standard deviation are still better than the other algorithms and for functions f5 and f6, although the improved GWO does not show the superiority of the algorithm, the difference with the other algorithms is not much. The improved GWO outperforms the other algorithms in terms of superiority-seeking ability and stability for the single-peak test function. The multi-peak test function is mainly used to test the exploration performance of the algorithm. The test results show that the improved GWO algorithm can reach the theoretical optimal value on f8 and f10, and f9. Although it cannot reach the optimal value, it is still better than other improved algorithms.

In summary, the improved GWO algorithm improves the performance of the 15 benchmark functions, and it is stable and robust, especially in f1–f4, f8, and f10. The improved algorithm can improve by several orders of magnitude, which is very obvious. The convergence speed of the improved GWO algorithm is also better than other improved algorithms, and during the experiment, it was found that the improved algorithm has excellent real-time performance and can effectively avoid the trap of local optimum in real-time, which proves the feasibility and superiority of the improved GWO algorithm compared with other improved algorithms.

#### 4.1.2. Convergence Speed Analysis

In order to visualize the convergence speed and search accuracy of the improved algorithm, the convergence curves of the analyzed 15 benchmark functions (d = 30) are shown in Figure 5. Figure 5a–e show the single-peak convergence curve, and (f–l) show the multi-peak convergence curve. Compared with several other algorithms, the convergence speed and search accuracy of the improved GWO algorithm is improved. The convergence curve verifies that the improved GWO algorithm solves single-peak and multi-peak functions. The improved algorithm in this paper can basically converge to the optimal value under the test of the benchmark function, and the last result is closer to the optimal value without acquired the best quality.

Moreover, it is found in the simulation process that the algorithm has good stability and a high success rate. The improved algorithm proposed in this paper has fewer iterations and higher optimal search accuracy than MGWO and NGWO, although they all can reach the optimal solution. Chaotic tent mapping, nonlinear convergence factor, and dynamic weighting strategy are combined in improved GWO, so that the problem of the algorithm falling into local optimum has been effectively solved and the convergence speed has been greatly improved. In summary, the improved algorithm can acquire a higher mean and standard deviation, which shows that the improved algorithm has higher solution accuracy and stability in most of the tested functions.

### 4.2. Comparison with Other Intelligent Optimization Algorithms

To further demonstrate the effectiveness of the improved algorithm, the improved algorithm is compared with the classical optimization algorithms Particle Swarm Optimization (PSO) algorithm, Sparrow Search Algorithm (SSA), and Mayfly Algorithm (MA) on 15 benchmark functions. The comparison results are shown in Table 4.

As can be seen from the results in Table 4, under the condition that the number of iterations is 500, compared with the other three classical algorithms, the improved GWO can reach the theoretical optimal value of 0 for the single-peaked benchmark functions f(1)–f(4), f(8), and f(10). In addition, the standard deviations and mean values got on the other benchmark functions have better performance, showing that the improved algorithm is practical and workable. The convergence curves are not put into the text due to length limitation. It is found that the improved algorithm has higher convergence accuracy and faster convergence speed by comparing the convergence curves with other intelligent algorithms.

Comparing algorithms based on mean and standard deviation values is not enough. Wilcoxon’s nonparametric statistical test is conducted at the 5% significance level to determine whether the improved GWO provides a significant improvement compared to other algorithms. The different algorithms on the benchmark function were employed to test the Wilcoxon rank-sum, and *P* and *R* values were obtained as a significant level indicator. If the *p* value is less than 0.05, the null hypothesis is rejected, and the two algorithms tested are considered significantly different. Conversely, the two algorithms tested are considered not to be significantly different. *R* result of ”+”, “−“, and “=“ represent, respectively, improved GWO performance better than, worse than, and equivalent to the comparison algorithm. If the *p* value is NaN, it means that the data is invalid, that is, the experimental results of the improved algorithm are similar to those of the compared algorithm, and their performance is similar.

This paper tests the Wilcoxon rank-sum with 30 repeated experiments on 15 benchmark functions by the improved GWO algorithm and other algorithms. The test results are shown in Table 5. In the most cases, the *R* values of the test results are “+”, except that the results *p* values for SSA, MA, and improved GWO on f5 are greater than 0.05 and the *R* values are “−”, and the results *p* values for MGWO and Improved GWO on f8 and f10 are NaN and the *R* values are “=”. This means the optimization efficiency of Improved GWO and MGWO is similar in f8 and f10. The results show that the Improved GWO algorithm’s performance is significantly improved compared with other algorithms in most cases.

### 4.3. Path Planning Application

#### 4.3.1. Problem Description

In path planning with obstacle avoidance for mobile robot, the mathematical model of robot environment should be established firstly replacing the virtual environment. After setting the start and end point of the mobile robot in the environment model, an intelligent algorithm is used to find a continuous curve that satisfies a specific performance index, which can avoid the obstacles in the environment.

The randomly generated individuals based on the intelligent optimization algorithm do not conform to the search space. It is necessary to establish a suitable fitness function consider various constraints, and then eliminate the individuals in the population who do not meet the constraints to acquire the better individuals. The mobile robot has to consider various factors in its actual operation. Therefore, it has the following main constraints.

Maximum cornering angle constraint

When using the algorithm for mobile robot path planning, it is necessary to consider the maximum steering angle constraint, which affects robot safety. This node is discarded if the specific rotation angle is outside the maximum performance range that the robot should withstand. If the rotation angle can satisfy the robot’s maneuverability, judge the other constraints. The maximum turning angle is specified as 60° in the simulation experiment.

2.Threat area constraints

Mobile robot path planning makes the robot reach its destination in the shortest distance while bypassing obstacles. The mathematical expression for the obstacle area can be got. Assuming that the distance between the robot and the center of the obstacle is *d_T_*, the damage to the robot caused by obstacle area, defined as Probability *P_T_*(*d_T_*), can be calculated as:(15)$$P_{T}(d_{T})=\left\{\begin{array}{c} 0,d_{T}>d_{T_{\max}}\\ \frac{1}{d_{T}},d_{T_{\min}}\leq d_{T}\leq d_{T_{\max}}\\ 1,d_{T}<d_{T_{\min}} \end{array}\right.$$
where *d_T_*_max_ indicates the maximum radius affected by the area, *d_T_*_min_ is the region where the probability of robot collision is 1.

#### 4.3.2. Path Planning

The main steps of applying the improved grey wolf algorithm to path planning are as follows:Establish the search space according to the actual environment, and set the starting point and target point.Initialize the parameters of grey wolf algorithm, including the number of wolves, the maximum number of iterations, tent mapping parameters, and upper and lower bounds for parameter values.Initialize the grey wolf’s position and objective function according to the utilization mapping.Calculate each grey wolf’s fitness and select the top three grey wolves as *wolf α*, *wolf β*, and *wolf w* for the fitness ranking.Compare with the objective function to update the position and the objective function.Update the convergence factor at each iteration.Calculate the next position of other wolves according to the positions of *wolf α*, *wolf β*, and *wolf w.*Reach the maximum number of iterations and output the optimal path.

To verify the performance of the improved GWO algorithm, the improved GWO algorithm applies to the path planning of mobile robot for verification analysis. The robot’s starting point is set as (0,0), and the target point is set as (100,100). The obstacles are generated randomly. a1 = 2, a2 = 0, the initial number of grey wolves is 30, and the maximum number of iterations is 500. the GWO, literature [10] MGWO, literature [11] NGWO, literature [12] GWO-fuzzy, literature [13] GWO-EPD, and the improved GWO algorithm in this paper, are applied to path planning for comparison. Figure 6a–e shows the obstacle avoidance paths planned by each improved GWO, and Figure 6f shows the convergence curves of the corresponding algorithms.

As shown in Figure 6a–e, except for MGWO, other improved algorithms find poorer and more costly paths. Although the path length of MGWO is short, the planned path is too close to the danger area, which is not conducive to the application of mobile robots. In addition, it can be seen from Figure 6f that the algorithm in this paper has better convergence compared with other improved algorithms. In summary, the improved GWO proposed in this paper can stably plan a safe path with optimal cost and satisfying constraints.

## 5. Conclusions

This paper proposes and applies an improved GWO to the path planning of mobile robot. First, an improved chaotic tent mapping is proposed, which is applied to the initial stage of the algorithm to increase the diversity of population initialization and improve the global search capability. Second, a nonlinear convergence factor based on the change curve of Gaussian distribution is used to balance the algorithm’s global search capability and local search capability. Finally, the traditional GWO is optimized with an improved dynamic weighting strategy. In order to test the competence of the improved GWO, 15 well-known benchmark functions having a wide range of dimensions and varied complexities are used in this paper. The results of the proposed improved GWO are compared to eight other algorithms. The results show that the improved GWO has better convergence speed and solution accuracy. In addition, the improved GWO is applied to the mobile robot path planning. The test results show that the improved GWO significantly improves cost consumption and convergence speed compared with other algorithms.

The improved GWO algorithm proposed in this paper is applied to mobile robots’ obstacle avoidance path planning. The situation of falling into local extremes can be avoided and the convergence speed and stability can be improved when the algorithm is applied to obstacle avoidance path planning of mobile robot. In the next research, we will continue to improve the algorithm and apply the improved algorithm to more practical mobile robots.

## Figures and Tables

**Figure 1 sensors-22-03810-f001:**
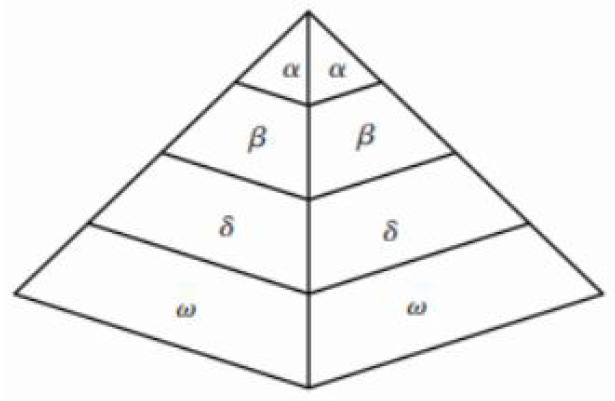
Grey wolf class system.

**Figure 2 sensors-22-03810-f002:**
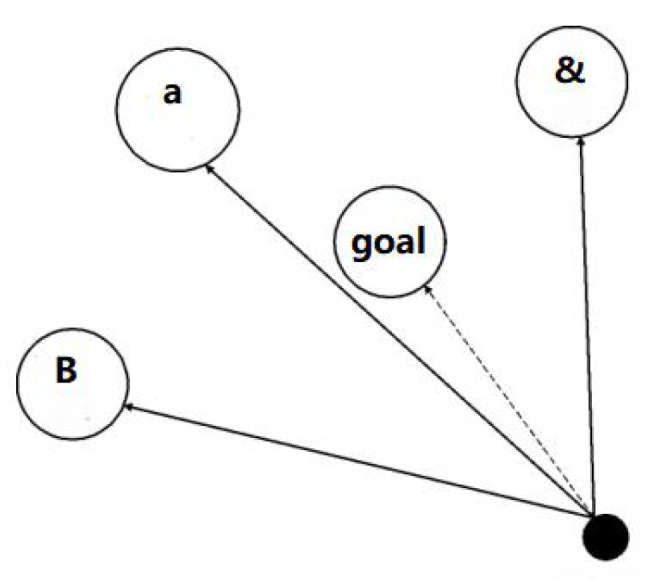
Prey tracing map.

**Figure 3 sensors-22-03810-f003:**
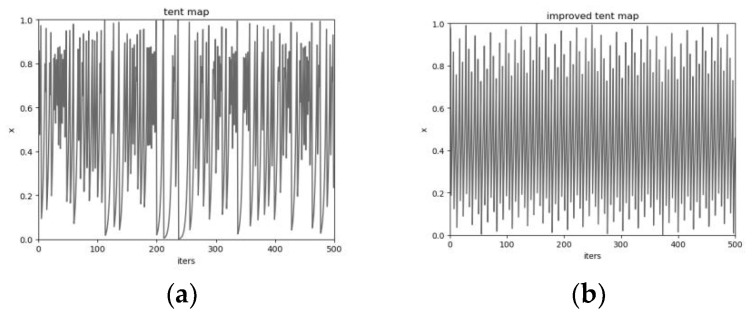
Chaotic mapping curve. (**a**) Tent; (**b**) improved tent.

**Figure 4 sensors-22-03810-f004:**
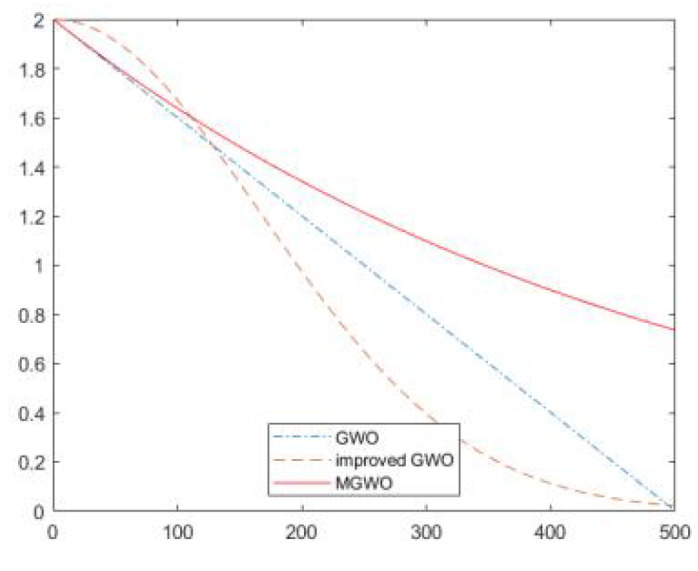
Convergence factor.

**Figure 5 sensors-22-03810-f005:**
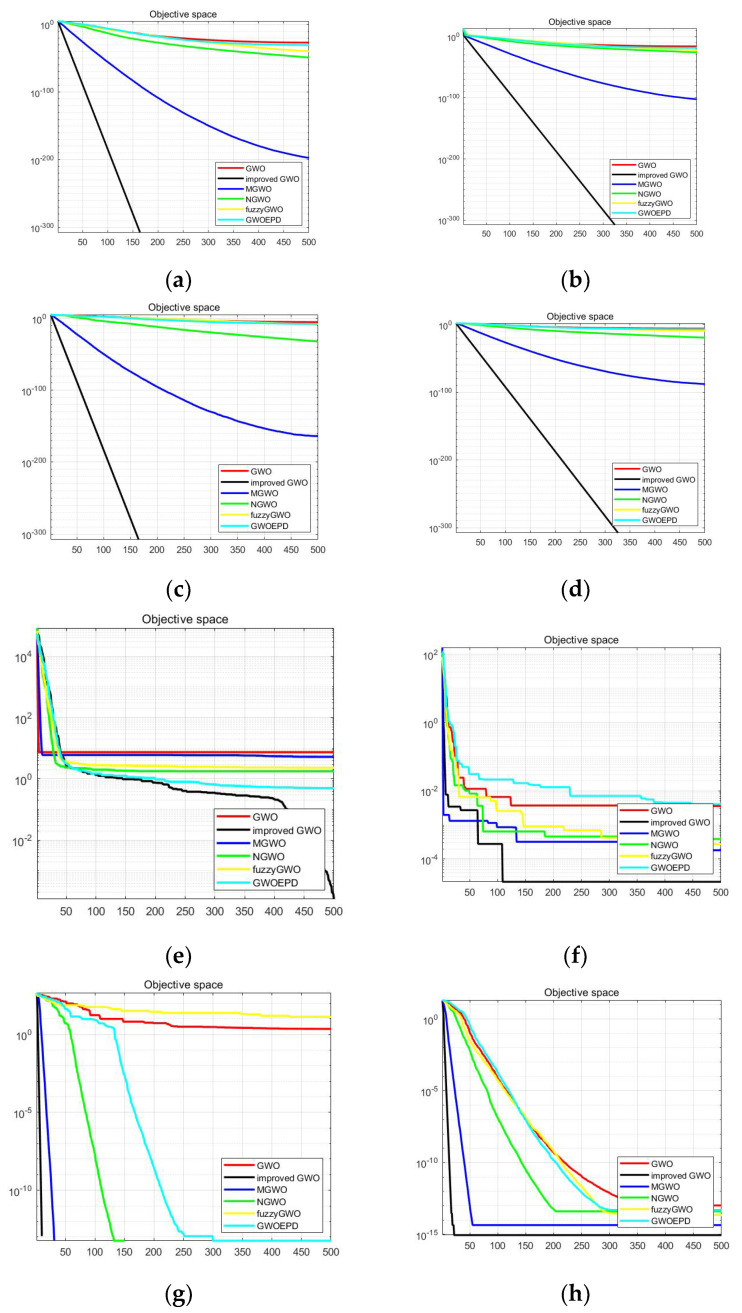
Convergence curves of algorithms on test function. (**a**) f1 function; (**b**) f2 function; (**c**) f3 function; (**d**) f4 function; (**e**) f5 function; (**f**) f7 function; (**g**) f8 function; (**h**) f9 function; (**i**) f10 function; (**j**) f11 function; (**k**) f12 function; (**l**) f14 function.

**Figure 6 sensors-22-03810-f006:**
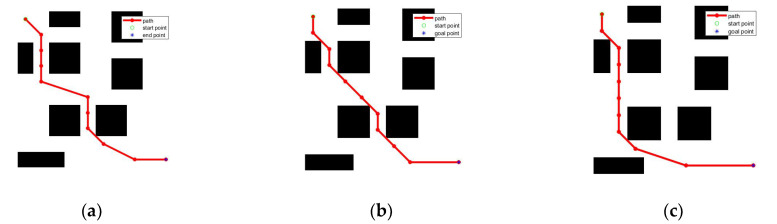
Path planning results. (**a**) Improved GWO; (**b**) MGWO; (**c**) NGWO; (**d**) GWO-fuzzy; (**e**) GWO-EPD; (**f**) convergence curves.

**Table 1 sensors-22-03810-t001:** Parameter Configuration.

Parameter Symbols	Meaning	Take Value
N	Population size	30
*T* _max_	Maximum Iteration	500
a1	Initial value of convergence factor	2
a2	Final value of convergence factor	0

**Table 2 sensors-22-03810-t002:** Benchmark functions.

Function	Dim	Scope	Solution
$f_{1}=\sum_{i=1}^{n}x_{i}^{2}$	30	[−100, 100]	0
$f_{2}=\sum_{i=1}^{n}x_{i}+\prod_{i=1}^{n}\left|x_{i}\right|$	30	[−10, 10]	0
$f_{3}={\sum_{i=1}^{n}\left(\sum_{j-1}^{i}x_{j}\right)}^{2}$	30	[−100, 100]	0
$f_{4}=\max_{i}\{|x_{i}|,1\leq i\leq n\}$	30	[−100, 100]	0
$f_{5}=\sum_{i=1}^{n-1}\left[100{\left(x_{i+1}-x_{i}^{2}\right)}^{2}+{\left(x_{i}-1\right)}^{2}\right]$	30	[−30, 30]	0
$f_{6}=\sum_{i=1}^{d}{\left(\left\lfloor x_{i}+0.5\right\rfloor \right)}^{2}$	30	[−100, 100]	0
$f_{7}=\sum_{i=1}^{n}ix_{i}^{4}+random[0,1)$	30	[−1.28, 1.28]	0
$f_{8}=\sum_{i=1}^{n}\left[x_{i}^{2}-10\cos(2\pi x_{i}^{})+10\right]$	30	[−5.12, 5.12]	0
$\begin{array}{l} f_{9}=-20\exp(-0.2\sqrt{\frac{1}{n}\sum_{i=1}^{n}x_{i}^{2}})\\ -\exp(\frac{1}{n}\sum_{i=1}^{n}\cos(2\pi x_{i}^{})+20+e) \end{array}$	30	[−32, 32]	0
$f_{10}=\frac{1}{4000}\sum_{i=1}^{n}x_{i}^{2}-\prod_{i=1}^{d}\cos(\frac{x_{i}}{\sqrt{i}})+1]$	30	[−600, 600]	0
$\begin{array}{l} f_{11}=\frac{\pi }{n}\{10\sin(\pi y_{1})+\sum_{i=1}^{D-1}{\left(y_{i}-1\right)}^{2}[1+10\sin^{2}(\pi y_{i+1})]+{\left(y_{n}-1\right)}^{2}\}\\ +\sum_{i=1}^{D}u(x_{i},5,100,4) \end{array}$	30	[−50, 50]	0.398
$\begin{array}{l} f_{12}=0.1\{10\sin(3\pi x_{1})+\sum_{i=1}^{D-1}{\left(x_{i}-1\right)}^{2}[1+10\sin^{2}(3\pi x_{i+1})]+{\left(x_{n}-1\right)}^{2}\}\\ +\sum_{i=1}^{D}u(x_{i},5,100,4) \end{array}$	30	[−50, 50]	3
$f_{13}=\sum_{i=1}^{D}\left|x_{i}^{}\sin(x_{i}^{})+0.1x_{i}^{}\right|$	30	[−10, 10]	0
$f_{14}=0.5+({\left(\sin(\sum_{i=1}^{D}x_{i}^{2})\right)}^{2}-0.5)\cdot {\left(1+0.001(\sum_{i=1}^{D}x_{i}^{2})\right)}^{-2}$	30	[−100, 100]	0
$f_{15}=(\sum_{i=1}^{D}\left[x_{i}^{2}+2x_{i+1}^{2}-0.3\cos(3\pi x_{i}^{})-0.4\cos(4\pi x_{i+1}^{})+0.7\right]$	30	[−15, 15]	0

**Table 3 sensors-22-03810-t003:** Test functions results.

Function	Algorithm	Average Value	Standard Deviation
f1	GWO	4.389 × 10^−27^	1.056 × 10^−27^
	Improved GWO	**0**	**0**
	MGWO	5.996 × 10^−199^	0
	NGWO	9.939 × 10^−49^	4.754 × 10^−48^
	GWO-fuzzy	9.887 × 10^−40^	4.977 × 10^−40^
	GWO-EPD	1.501 × 10^−31^	2.289 × 10^−30^
f2	GWO	2.167 × 10^−5^	3.958 × 10^−6^
	Improved GWO	**0**	**0**
	MGWO	1.617 × 10^−102^	2.154 × 10^−102^
	NGWO	2.133 × 10^−26^	1.143 × 10^−26^
	GWO-fuzzy	1.572 × 10^−24^	1.374 × 10^−23^
	GWO-EPD	1.893 × 10^−19^	2.358 × 10^−20^
f3	GWO	1.115 × 10^−7^	3.463 × 10^−5^
	Improved GWO	**0**	**0**
	MGWO	6.982 × 10^−166^	0
	NGWO	1.015 × 10^−33^	3.789 × 10^−31^
	GWO-fuzzy	5.981 × 10^−8^	3.753 × 10^−7^
	GWO-EPD	4.505 × 10^−8^	2.456 × 10^−6^
f4	GWO	8.423 × 10^−7^	4.583 × 10^−7^
	Improved GWO	**0**	**0**
	MGWO	5.368 × 10^−90^	9.664 × 10^−89^
	NGWO	4.414 × 10^−20^	1.104 × 10^−19^
	GWO-fuzzy	4.995 × 10^−9^	8.259 × 10^−7^
	GWO-EPD	3.395 × 10^−7^	7.652 × 10^−6^
f5	GWO	2.706 × 10^1^	6.824 × 10^−1^
	Improved GWO	2.867 × 10^1^	2.611 × 10^−2^
	MGWO	2.761 × 10^1^	3.917 × 10^−1^
	NGWO	2.719 × 10^1^	5.836 × 10^−1^
	GWO-fuzzy	2.855 × 10^1^	8.518 × 10^−1^
	GWO-EPD	2.818 × 10^1^	8.075 × 10^−1^
f6	GWO	1.013	2.816 × 10^−1^
	Improved GWO	6.533 × 10^−1^	2.860 × 10^−1^
	MGWO	5.261	6.381 × 10^−1^
	NGWO	1.829	3.763 × 10^−1^
	GWO-fuzzy	2.324	5.052 × 10^−1^
	GWO-EPD	1.238	4.725 × 10^−1^
f7	GWO	1.154 × 10^−3^	1.226 × 10^−3^
	Improved GWO	**2.961** × 10^−7^	**2.373** × 10^−7^
	MGWO	1.914 × 10^−4^	1.369 × 10^−4^
	NGWO	1.347 × 10^−3^	2.747 × 10^−4^
	GWO-fuzzy	1.744 × 10^−3^	1.047 × 10^−3^
	GWO-EPD	1.646 × 10^−3^	1.031 × 10^−3^
f8	GWO	6.934 × 10^−12^	4.701
	Improved GWO	**0**	**0**
	MGWO	0	0
	NGWO	5.684 × 10^−14^	2.017 × 10^−1^
	GWO-fuzzy	6.130 × 10^−1^	1.657 × 10^−1^
	GWO-EPD	1.715 × 10^−13^	3.852
f9	GWO	1.103 × 10^−13^	1.633 × 10^−14^
	Improved GWO	**8.811** × 10^−16^	**1.164** × 10^−16^
	MGWO	4.440 × 10^−15^	6.486 × 10^−15^
	NGWO	2.930 × 10^−14^	2.420 × 10^−15^
	GWO-fuzzy	2.930 × 10^−14^	3.923 × 10^−15^
	GWO-EPD	4.352 × 10^−14^	6.4963 × 10^−15^
f10	GWO	7.558 × 10^−3^	1.412 × 10^−2^
	Improved GWO	**0**	**0**
	MGWO	0	0
	NGWO	0	0
	GWO-fuzzy	7.2159 × 10^−4^	3.0047 × 10^−3^
	GWO-EPD	5.6751 × 10^−3^	5.7892 × 10^−3^
f11	GWO	3.8124 × 10^−1^	6.7824 × 10^−2^
	Improved GWO	2.1331 × 10^−3^	6.8945 × 10^−3^
	MGWO	5.3122 × 10^−1^	3.1121 × 10^−2^
	NGWO	1.1021 × 10^1^	3.0031
	GWO-fuzzy	1.3811	8.3221
	GWO-EPD	1.2254 × 10^−2^	4.2214 × 10^−1^
f12	GWO	7.3712	4.1077 × 10^−1^
	Improved GWO	1.2922 × 10^−2^	7.6012 × 10^−2^
	MGWO	8.3211	3.2454 × 10^−1^
	NGWO	1.6722 × 10^1^	3.1207
	GWO-fuzzy	6.1545 × 10^−1^	4.5512
	GWO-EPD	8.21475 × 10^2^	8.1542 × 10^2^
f13	GWO	4.5214 × 10^−3^	2.5784 × 10^−3^
	Improved GWO	2.4457 × 10^−6^	6.3641 × 10^−6^
	MGWO	7.7541 × 10^−5^	8.2231 × 10^−4^
	NGWO	2.1441 × 10^1^	8.1601
	GWO-fuzzy	1.2215 × 10^1^	2.2232 × 10^1^
	GWO-EPD	1.2014 × 10^−2^	1.2424 × 10^1^
f14	GWO	1.4125 × 10^−2^	2.3622 × 10^−3^
	Improved GWO	3.1337 × 10^−3^	1.1184 × 10^−3^
	MGWO	4.3221 × 10^−3^	1.4752 × 10^−3^
	NGWO	4.8842 × 10^−1^	2.4821 × 10^−3^
	GWO-fuzzy	1.3315 × 10^−2^	2.4774 × 10^−1^
	GWO-EPD	3.9454 × 10^−1^	1.7424 × 10^−1^
f15	GWO	1.2547 × 10^−10^	7.2242 × 10^−11^
	Improved GWO	2.4467 × 10^−13^	1.0871 × 10^−14^
	MGWO	7.2101 × 10^−4^	7.9945 × 10^−5^
	NGWO	1.5547 × 10^1^	9.0141
	GWO-fuzzy	2.4875 × 10^−13^	1.0401 × 10^1^
	GWO-EPD	7.2154 × 10^2^	9.4012 × 10^1^

**Table 4 sensors-22-03810-t004:** Test functions results.

Function	Algorithm	Average Value	Standard Deviation
f1	Improved GWO	**0**	**0**
	PSO	3.125 × 10^−2^	2.716 × 10^−2^
	SSA	1.891 × 10^−257^	0
	MA	1.711 × 10^−43^	4.254 × 10^−43^
f2	Improved GWO	**0**	**0**
	PSO	1.416 × 10^−1^	3.581^−1^
	SSA	1.435 × 10^−93^	8.487 × 10−9^3^
	MA	2.255 × 10^2^	8.183 × 10^2^
f3	Improved GWO	**0**	**0**
	PSO	7.225 × 10^−2^	5.331 × 10^−1^
	SSA	2.821 × 10^−180^	0
	MA	7.318 × 10^−5^	5149 × 10^−4^
f4	Improved GWO	**0**	**0**
	PSO	9.225 × 10^−2^	1.153 × 10^−1^
	SSA	1.354 × 10^−93^	6.81 × 10^−93^
	MA	8.154 × 10^−7^	6.518 × 10^−5^
f5	Improved GWO	2.867 × 10^1^	2.611 × 10^−2^
	PSO	1.314 × 10^2^	1.795 × 10^2^
	SSA	2.327 × 10^−3^	2.189 × 10^−3^
	MA	4.501 × 10^−1^	5.587 × 10^−1^
f6	Improved GWO	6.533	2.801 × 10^−1^
	PSO	8.792 × 10^5^	9.782 × 10^5^
	SSA	1.047 × 10^1^	4.772
	MA	3.128 × 10^1^	8.791 × 10^2^
f7	Improved GWO	**2.961** × 10^−7^	**2.373** × 10^−7^
	PSO	2.561 × 10^−1^	7.844 × 10^−1^
	SSA	1.144 × 10^−4^	3.581 × 10^−3^
	MA	3.254 × 10^−2^	4.358 × 10^−1^
f8	Improved GWO	**0**	**0**
	PSO	3.015	2.641
	SSA	8.161 × 10^−185^	1.254 × 10^−186^
	MA	2.271 × 10^−45^	5.174 × 10^−44^
f9	Improved GWO	**8.881** × 10^−16^	**1.604** × 10^−16^
	PSO	3.712 × 10^−2^	2.816 × 10^−1^
	SSA	8.881 × 10^−16^	0
	MA	4.213 × 10^−10^	1.576 × 10^−9^
f10	Improved GWO	**0**	**0**
	PSO	5.001 × 10^−3^	2.655 × 10^−1^
	SSA	4.114 × 10^−210^	3.241 × 10^−211^
	MA	5.260 × 10^−140^	0
f11	Improved GWO	2.1331 × 10^−3^	6.8945 × 10^−3^
	PSO	1.8741	4.4411
	SSA	1.496 × 10^−2^	2.106 × 10^−2^
	MA	2.714 × 10^−1^	1.954 × 10^−17^
f12	Improved GWO	1.292 × 10^−2^	7.6012 × 10^−2^
	PSO	8.4152	8.3372
	SSA	7.346 × 10^−1^	1.355 × 10^−2^
	MA	8.214	1.245 × 10^−2^
f13	Improved GWO	2.4457 × 10^−6^	6.3641 × 10^−6^
	PSO	1.052 × 10^2^	1.2362
	SSA	1.232 × 10^−3^	1.571 × 10^−4^
	MA	3.247 × 10^−3^	5.014 × 10^−3^
f14	Improved GWO	3.1337 × 10^−3^	1.1184 × 10^−3^
	PSO	3.958 × 10^−1^	1.541 × 10^−2^
	SSA	9.001 × 10^−2^	0
	MA	3.971 × 10^−1^	6.051 × 10^−1^
f15	Improved GWO	2.4467 × 10^−13^	1.0871 × 10^−14^
	PSO	7.1522	9.142 × 10^1^
	SSA	4.701 × 10^−7^	3.147 × 10^−8^
	MA	5.445 × 10^−2^	4.401 × 10^−2^

**Table 5 sensors-22-03810-t005:** Wilcoxon’s rank test of Improved GWO and other algorithms on 15 benchmark functions.

Function		GWO	MGWO	NGWO	GWO-Fuzzy	GWO-EPD	SSA	MA	PSO
**f1**	P	6.52 × 10^−^^12^	8.78 × 10^−^^8^	5.05 × 10^−^^12^	6.52 × 10^−^^12^	6.52 × 10^−^^12^	6.01 × 10^−^^5^	6.52 × 10^−^^12^	6.52 × 10^−^^12^
	R	+	+	+	+	+	+	+	+
f2	P	2.07 × 10^−^^11^	1.40 × 10^−^^11^	2.07 × 10^−^^11^	2.07 × 10^−^^11^	2.07 × 10^−^^11^	2.07 × 10^−^^11^	2.07 × 10^−^^11^	2.07 × 10^−^^11^
	R	+	+	+	+	+	+	+	+
f3	P	3.77 × 10^−^^10^	6.52 × 10^−^^12^	6.52 × 10^−^^12^	6.52 × 10^−^^12^	6.52 × 10^−^^12^	3.77 × 10^−^^10^	6.52 × 10^−^^12^	6.52 × 10^−^^12^
	R	+	+	+	+	+	+	+	+
f4	P	6.52 × 10^−^^12^	5.05 × 10^−^^11^	6.52 × 10^−^^12^	6.52 × 10^−^^12^	6.52 × 10^−^^12^	3.77 × 10^−^^11^	6.52 × 10^−^^12^	6.52 × 10^−^^12^
	R	+	+	+	+	+	+	+	+
f5	P	4.60 × 10^−^^3^	1.20 × 10^−5^	6.01 × 10^−3^	1.09 × 10^−2^	1.68 × 10^−4^	2.05 × 10^−2^	4.23 × 10^−1^	1.20 × 10^−6^
	R	+	+	+	+	+	-	-	+
f6	P	2.07 × 10^−^^11^	1.41 × 10^−11^	2.07 × 10^−^^11^	2.07 × 10^−^^11^	2.07 × 10^−^^11^	2.07 × 10^−^^11^	2.07 × 10^−^^11^	2.07 × 10^−^^11^
	R	+	+	+	+	+	+	+	+
f7	P	3.01 × 10^−11^	5.24 × 10^−9^	3.01 × 10^−11^	2.07 × 10^−^^11^	2.07 × 10^−^^11^	2.07 × 10^−^^11^	2.07 × 10^−^^11^	2.07 × 10^−^^11^
	R	+	+	+	+	+	+	+	+
f8	P	6.52 × 10^−^^12^	NaN	6.52 × 10^−^^12^	6.52 × 10^−^^12^	6.52 × 10^−^^12^	2.07 × 10−11	6.52 × 10^−^^12^	6.52 × 10^−^^12^
	R	+	=	+	+	+	+	+	+
f9	P	2.07 × 10^−^^11^	2.07 × 10^−^^11^	2.07 × 10^−^^11^	2.07 × 10^−^^11^	2.07 × 10^−^^11^	3.77 × 10^−10^	2.07 × 10^−^^11^	2.07 × 10^−^^11^
	R	+	+	+	+	+	+	+	+
f10	P	6.52 × 10^−^^12^	NaN	NaN	6.52 × 10^−^^12^	6.52 × 10^−^^12^	2.07 × 10^−^^11^	2.07 × 10^−^^11^	6.52 × 10^−^^12^
	R	+	=	=	+	+	+	+	+
f11	P	6.52 × 10^−^^12^	2.07 × 10^−^^11^	2.07 × 10^−^^11^	2.07 × 10^−^^11^	6.52 × 10^−^^12^	2.07 × 10^−^^11^	2.07 × 10^−^^11^	6.52 × 10^−^^12^
	R	+	+	+	+	+	+	+	+
f12	P	6.52 × 10^−^^12^	2.07 × 10^−^^11^	2.07 × 10^−^^11^	2.07 × 10^−^^11^	6.52 × 10^−^^12^	2.07 × 10^−^^11^	2.07 × 10^−^^11^	6.52 × 10^−^^12^
	R	+	+	+	+	+	+	+	+
f13	P	6.52 × 10^−^^12^	1.20e−06	6.52 × 10^−^^12^	6.52 × 10^−^^12^	2.07 × 10^−^^11^	2.07 × 10^−^^11^	2.07 × 10^−^^11^	6.52 × 10^−^^12^
	R	+	+	+	+	+	+	+	+
f14	P	6.52 × 10^−^^12^	2.07 × 10^−^^11^	2.07 × 10^−^^11^	2.07 × 10^−^^11^	2.07 × 10^−^^11^	2.07 × 10^−^^11^	2.07 × 10^−^^11^	6.52 × 10^−^^12^
	R	+	+	+	+	+	+	+	+
f15	P	2.07 × 10^−^^11^	6.52 × 10^−^^12^	6.52 × 10^−^^12^	2.07 × 10^−^^11^	6.52 × 10^−^^12^	6.52 × 10^−^^12^	6.52 × 10^−^^12^	6.52 × 10^−^^12^
	R	+	+	+	+	+	+	+	+

## Data Availability

The processed data required to reproduce these findings cannot be shared as the data also forms part of an ongoing study.

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
