# Peer review of "Improved Grey Wolf Optimization Algorithm and Application"

_sensors, 2022, doi:10.3390/s22103810_

Round 1

Reviewer 1 Report

Title of the paper: "Improved Grey Wolf Optimization Algorithm and Application"

    As a researcher working in the same field, I am impressed by the technique introduced in the paper, because it sheds new light on the earlier results of several authors and obviously can be successfully used in practice. From this point of view, the subject of the paper fits well with the scope of the journal (Sensors).

The paper is ended with numerical simulations that corroborate the theoretical results.

This manuscript contains new ideas and good results that help other researchers.

The decision is too major revision for publication in the "Sensors".

Therefore, I recommend publishing this work after taking these points into account.

1-The English writing of the paper is required to be improved. Please check the manuscript carefully for typos and grammatical errors. I found some typos and grammatical errors within this manuscript, which have been excluded from my review. In addition, the English structure of the article, including punctuation, semicolon, and other structures, must be carefully reviewed.

2-In the introduction, the authors did not provide a strong motivation for the paper and the obtained results. In addition, they should discuss the main contributions of their work in detail after the motivation part. Then they should summarize the main structure of their paper in brief at the end of the introduction.

3-The literature review about the problem under study is not adequate. I suggest the authors keep up-to-date the introductory part by the recent relevant developments and publications.

4-I found no comparative results within this manuscript. Some comparative results with the other methods available in the literature would be expected in the revision.

5-The introduction needs to be improved by the recent developments in the field of numerical simulation and stability as well as its applications. For this purpose, the authors can add the following references to enrich the introductory section:

*Analytical solution of magneto-photothermal theory during variable thermal conductivity of a  semiconductor material due to pulse heat flux and volumetric heat source, Waves in Random and Complex Media, vol. 31, no. 6, pp. 2040–2057, 2021.

6-Future recommendations should be added to assist other researchers to extend the presented research analysis.

Reviewer 2 Report

The paper presents an improved GWO algorithm for boosting the performance and accuracy of GWO. The paper needs rigorous review before acceptance. The authors are invited to conduct the following modifications before acceptance as:

1- The main problem of this paper is poor benchmarking. The authors have used 10 benchmark functions to benchmark the algorithm which are not very significant numerical optimization problems. The authors are suggested to follow the recent CEC benchmark sets.
2- The authors can note that the results in Table 4 are very similar and can be statistically insignificant. Please conduct a non-parametric study for the obtained results. The authors can refer to the Wilcoxon rank test as an example.

3- The results in Table 3 suffer from similar problems. The authors are requested to do the same as above.  

4- The comparison with other metaheuristics is relatively poor. Further comparison is required with other state-of-the-art algorithms.  

5- The discussion in the present form is relatively weak and should be strengthened with more details and justifications. A separate discussion section should be added to the manuscript.

6-The contributions and novelty of the paper should be added before the last paragraph in section 1.

7-Both abstract and conclusion should be written in a more scientific way.

8- The main application of this paper which is path planning is very poorly discussed. Further theoretical background and mathematical description should be added, as well as a comparison with other state-of-the-art algorithms. 

Reviewer 3 Report

The paper presents an improved grey wolf optimization algorithm to solve the problems of low accuracy and slow convergence of the grey wolf optimization algorithm. The paper is suitable for the Sensors journal, but it needs extensive edition and language corrections as it is very poorly written. Moreover, the paper's novelty is not well explained and is unclear. In the introduction, please explain if the GWO improvements are proposed by you solely or if other similar works inspire you. Please also provide quantitative measures for the "convergence accuracy" and "convergence speed" of the new algorithm compared to other tested algorithms. Other my comments:

  • Abstract, introduction and main text. Please use "." instead of ";".
  • Line 8, 97, 209, 210: the space is missing.
  • Line 25-27 – sentence has some spelling and syntax mistakes.
  • Line 50, 64: when referring, do not use first name initials.
  • Line 64: the reference number is missing.
  • Line 73: please use the Greek symbol "omega".
  • Line 92: "Eq" is redundant.
  • 2 needs improvements. Symbols should be in Greek letters. The same problem is in lines: 107-109, 125, 194, 333, and many others.
  • Line 109: instead of " (6):", use "where:".
  • Line 109: instead of "Eq. (7):", write "The eq. (7) presents…".
  • Line 123: abbreviation "PWLCM" was not explained. Similarly, in lines 299 and 300 some abbreviations were not explained.
  • I suggest writing the names like: "logistic mapping", "tent mapping", "tent falling", etc., using minor letters.
  • (9): "=" is missing.
  • Symbol "N" in eq. (9) is not explained.
  • Line 145: "lb", "ub" should be written using italic style. I found many similar mistakes in the paper.
  • Line 169, 171: do not explain symbols that were explained previously.The same is in lines 314 and 315.
  • Line 171: something went wrong.
  • "The improved GWO algorithm pseudo-code is as follows:" – such form is not acceptable. Refer to Table 1 in the text.
  • Table 3: unknown numbers "2.22", "1.2", "2.21"
  • Line 251: "F8" or f8?
  • 5 and Fig. 6b are completely illegible. Lines are not visible. The descriptions are too small.

Round 2

Reviewer 1 Report

Dear Prof. Dr. (Editor in Sensors), 
Thank you very much the editor of this journal for giving me the chance to consider such a valuable paper.

The revised version has been modified according to my remarks. It seems to be right. 
Ref. 17 is (Waves in Random and Complex Media 31 (6), 2040-2057‏). 
Also, can be added the Ref.
Thermal-piezoelectric problem of a semiconductor medium during photo-thermal excitation, Waves in Random and Complex Media 31 (6), 2499-2513.

Then, it can be published.
Thank you very much.
Prof. Dr. Amr Mahdy

Author Response

Dear Editors and Reviewers:

Thank you for your letter and the reviewers’ comments concerning our manuscript entitled “Improved Grey Wolf Optimization Algorithm and Application”. Those comments are valuable and helpful for revising and improving our paper and the essential guiding significance to our research. We have studied the comments carefully and have made a correction which we hope meets with approval(the revised parts of the text are marked in red font). The significant corrections in the paper and the responses to the reviewer’s comments are as follows:

Point 1:Ref. 17 is (Waves in Random and Complex Media 31 (6), 2040-2057‏). 
Also, can be added the Ref.
Thermal-piezoelectric problem of a semiconductor medium during photo-thermal excitation, Waves in Random and Complex Media 31 (6), 2499-2513.

Response 1: According to the comment, related content has been improved: After learning about the recent advances in the application of GWO algorithms to multiple fields, I was particularly inspired by the recent developments in numerical simulation and stability and their applications. The introduction section and the references section were later reworked. The details of the revision are given in the third paragraph of the introduction and references [17], [18].

Reviewer 2 Report

The authors have modified their paper according to my comments. Even though the paper can be further improved, the experiments show the efficiency of the algorithm. Therefore, the paper can be accepted for publication in Sensors. 

Author Response

Dear Editor and Reviewers.
       Thank you for your letter and the reviewers' acceptance of our manuscript entitled "Improved Gray Wolf Optimization Algorithm and Applications".

Reviewer 3 Report

The authors improved the paper in-line with my suggestions. Therefore, it may be accepted for publication.

Author Response

(The authors gave the same response as above.)
